# The Patterns and Appropriateness of Systemic Antifungal Prescriptions in a Regional Hospital in Hong Kong

**DOI:** 10.3390/antibiotics14060556

**Published:** 2025-05-29

**Authors:** Ryan Y. H. Leung, Jimmy Y. W. Lam

**Affiliations:** Department of Pathology, Pamela Youde Nethersole Eastern Hospital, Hong Kong SAR, China; lyw543a@ha.org.hk

**Keywords:** invasive fungal diseases, antifungal therapy, antimicrobial stewardship

## Abstract

Introduction: The consumption of systemic antifungals is on the rise. However, a significant proportion of systemic antifungal prescriptions is inappropriate. Inappropriately prescribed antifungals are problematic, but there has been minimal emphasis on ensuring the appropriate prescription of systemic antifungals. Local studies regarding the patterns and appropriateness of antifungal prescriptions are also lacking. Materials and Methods: In this retrospective, single-centre, observational study, every in-patient prescription order of systemic antifungals in a regional hospital in Hong Kong between 1 May and 31 July 2023 was reviewed via electronic patient records. The appropriateness of a systemic antifungal prescription was assessed by its indication, dosage, duration and antifungal–concomitant drug interactions by a single reviewer. Results: A total of 177 prescriptions orders were collected. Itraconazole, micafungin and fluconazole were the most prescribed systemic antifungals. The haematology team, infectious disease team and ICU were the major systemic antifungal prescribers in this study. The overall appropriateness of systemic antifungal prescriptions was 27.7% (49/177), with an appropriateness of 72.9% (129/177) for indications, 57.1% (101/177) for dosage, 91.5% (162/177) for duration and 71.6% (127/177) for antifungal–concomitant drug interactions. Triazole antifungals had an overall prescription appropriateness of only 15% and were more likely to be prescribed inappropriately than non-triazole antifungals (*p* < 0.001). Common prescription pitfalls include (i) starting a systemic antifungal for sputum culture that grew *Candida* spp., (ii) debatable prophylaxis with itraconazole capsules, (iii) overlooking potentially serious antifungal–drug interactions. Conclusions: Inappropriate systemic antifungal prescription is not uncommon in Hong Kong. Establishing an antifungal stewardship programme in public hospitals may be beneficial.

## 1. Introduction

With advances in medicine, we have an ever-growing population susceptible to invasive fungal disease (IFD) due to old age and underlying conditions such as haematological malignancy, chemotherapy, medication-induced immunosuppression, HIV/AIDS, intra-abdominal surgery and total parenteral nutrition. As a result, there has been an increase in the incidence of IFD and a subsequently increased use of systemic antifungals in the last two decades [1,2]. However, a significant proportion of systemic antifungal prescriptions is inappropriate; published studies have reported that the appropriateness of systemic antifungal prescriptions ranges from 29.4% to 74.8% [3,4,5,6]. Knowledge of systemic antifungal prescriptions and IFD diagnosis and management among healthcare providers, in general, is suboptimal at best [7,8], which has largely contributed to the widespread inappropriate use of antifungals [9].

Patients who receive inappropriate antifungal treatment are associated with higher mortality, in addition to longer lengths of stay in hospital and increased hospitalisation costs [10]. As with other antimicrobial agents, the misuse and overuse of antifungal agents exert positive selective pressure for resistant fungal pathogens [11,12,13]. Although not as common as antibiotic resistance in clinical practice, fungal pathogens with intrinsic resistance (e.g., *Candida krusei* to fluconazole) and acquired resistance (e.g., *Aspergillus fumigatus* with TR34/L98H mutation in CYP51A gene to triazoles) to antifungals pose a significant challenge to IFD management, as their treatment options are often limited to a few of antifungals [3]. The widespread use of immunosuppressants and overdiagnosis of a loosely defined COVID-19-associated pulmonary aspergillosis (CAPA) in the recent COVID-19 pandemic raises further uncertainties in antifungal prescription appropriateness in an era of defensive medicine [14]. A multi-centre retrospective study in 2022 noted significant increase in voriconazole and caspofungin prescriptions at four medical sites with 14 intensive care units during the COVID-19 pandemic [15]. The emergence of fluconazole-resistant *Candida parapsilosis* during the COVID-19 pandemic and ongoing outbreaks of *Candida auris* in Hong Kong since 2020 further highlight the need for better understandings of antifungal prescription patterns and the appropriateness of antifungal prescriptions in Hong Kong [16].

However, minimal emphasis has been placed on ensuring the appropriate prescription of systemic antifungals [3]; this is highlighted by the scarcity of antifungal stewardship programmes and local studies evaluating antifungal prescribing patterns and appropriateness. Reliable reports of antifungal consumption data are also lacking [17].

Our study is designed to address the aforementioned knowledge gap and provide insights on (i) antifungal prescription patterns, (ii) the appropriateness of systemic antifungal prescriptions, (iii) common pitfalls of antifungal prescriptions, (iv) how to improve the quality of antifungal prescription in the public hospitals of Hong Kong.

## 2. Materials and Methods

### 2.1. Study Design

This retrospective, observational study was conducted at Pamela Youde Nethersole Eastern Hospital (PYNEH) in 2024. PYNEH is an 1884-bed, major acute hospital that provides 24 h emergency and comprehensive secondary service and tertiary services for the Eastern District of Hong Kong Island. PYNEH houses an Intensive Care Unit (ICU) (25 beds), a clinical oncology unit (66 beds) and a haematology unit (20 beds), among many other specialties.

All hospitalised patients in PYNEH who were prescribed an oral or a parenteral antifungal agent between 1 May and 31 July 2023 were included in this study. Patients who were prescribed topical or vaginal antifungals were excluded. The following systemic antifungals were available in PYNEH and were included in this study:(i)Triazoles: fluconazole, itraconazole, voriconazole, isavuconazole and posaconazole;(ii)Polyenes: amphotericin B deoxycholate and liposomal amphotericin B;(iii)Echinocandins: anidulafungin and micafungin;(iv)Squalene epoxidase inhibitor: terbinafine;(v)Anti-metabolite: flucytosine.

### 2.2. Ethics Approval

This research was conducted in accordance with the Declaration of Helsinki and institutional standards. Ethics approval was obtained from the Hospital Authority Central Institutional Review Board (Ref: CIRB-2024-030-4).

### 2.3. Data Collection

A retrospective chart review was conducted through electronic patient records (EPRs) to collect the following data: (i) patient demographics and clinical characteristics (e.g., age, sex, underlying medical conditions), (ii) fungal disease characteristics (e.g., working diagnosis, clinical and radiological evidence of fungal infection, mycological culture and indirect mycological tests and clinical outcome), (iii) antifungal therapy (e.g., prescribing specialty, indication, dosage, duration of antifungals and antifungal–concomitant drug interaction).

### 2.4. Antifungal Therapeutic Strategies

Antifungal prescriptions were classified into prophylactic, empirical, pre-emptive or targeted therapy using a modified version of the classification proposed by Herbrecht and Berceanu [18] (Appendix A). Modifications were made to accommodate patients that did not have leukaemia or haematological malignancy. The classification of invasive fungal disease per EORTC/MSG criteria [19] was also incorporated into our classification.

### 2.5. Evaluation of Appropriateness of Antifungal Prescription

The appropriateness of each antifungal prescription was based on assessment criteria modified from the criteria previously described by Nivoix et al. [4] (Appendix A). Indication, dosage, duration and potential antifungal–concomitant drug interactions were evaluated. An antifungal prescription was deemed appropriate when all four evaluation criteria were met, debatable when there was at least one debatable assessment criterion without any inappropriate assessment criteria and inappropriate when there was at least one inappropriate assessment criterion. Dr. Ryan Leung (the author), who is a member of the microbiology team, was the single reviewer that evaluated the appropriateness of each systemic antifungal prescription. The study was carried out under the auspices of the microbiology team.

Indications for antifungals and choice of antifungals were assessed based on the recommendations made by the Infectious Disease Society of America (IDSA) [20,21,22,23,24,25], European Conference on Infections in Leukaemia (ECIL) [26,27] and other published studies [28,29,30]. Dosage was assessed according to the summary of product characteristics (SPC) with respect to IDSA or ECIL guidelines, liver and renal function, the presence or absence of loading dose, dosage form and the use of therapeutic drug monitoring (TDM) whenever possible. The duration of treatment was assessed according to the instruction of the IDSA, ECIL and the literature. The potential for antifungal–concomitant drug interactions and their risk categories were assessed using the UpToDate Lexidrug online software (https://www.uptodate.com/drug-interactions/#di-druglist, accessed on 28 June 2024; Lexi-Comp Inc., Hudson, OH, USA).

### 2.6. Statistical Analysis

Anonymized data were first tabulated into a relational database (Microsoft Excel 365, Seattle, WA, USA). Descriptive statistics were used to analyse the data collected in this study. Frequencies and percentages were recorded for categorical variables, whereas the mean (or median) and standard deviation (or interquartile range) were reported for continuous variables. The chi-square test for independence was performed to examine the relationships of antifungal classes and prescription appropriateness.

## 3. Results

A total of 177 systemic antifungal prescriptions were made between 1 May and 31 July 2023 for 146 hospital admissions in 125 patients. Multiple systemic antifungals were prescribed in 31 (21.2%) hospital admissions. Among the 146 hospital admissions in which patients were prescribed a systemic antifungal, 107 (73.7%) patients survived the hospitalisation episode and 39 patients (26.7%) passed away.

### 3.1. Patient Characteristics

A total of 125 patients were included in this study, of whom 50.4% (*n* = 63) were male. The mean age of the study cohort was 67.1 years old (range: 0 days–103 years).

Among the 177 systemic antifungal prescriptions, 95 (53.7%) were given to patients with haematological malignancy, 86 (48.6%) were given to patients with recent chemotherapy, 78 (44.1%) were given to patients who had prolonged and severe neutropenia (defined as absolute neutrophil count < 500 cells/uL and >7 days), 49 (27.7%) were given to patients with recent corticosteroid therapy, 28 (15.8%) were given to patients with COVID-19 infection, 27 were given to (15.3%) patients with a central venous catheter, 24 (13.6%) were given to patients with diabetes mellitus and 24 (13.6%) were given to patients on other immunosuppressants (e.g., tacrolimus and mycophenolate) (Table 1).

### 3.2. Clinical Characteristics

Radiological evidence of IFD was noted in 11 out of 177 (6.2%) systemic antifungal prescriptions. A recent positive fungal culture was reported in 55 out of 177 (31.1%) systemic antifungal prescriptions. A total of 61 fungal organisms were isolated, of which 51 were yeast (92.7%), 8 were mould (14.5%) and 2 were dimorphic fungi (3.6%). *Candida albicans* (*n* = 19; 34.5%) was the most common yeast isolated, which was followed by *Candida glabrata* (*n* = 9; 16.4%) and *Candida parapsilosis* (*n* = 8; 14.5%). *Aspergillus fumigatus* (*n* = 5; 9.1%) was the most common mould isolated, whereas *Talaromyces marneffeii* (*n* = 2; 3.6%) was the sole species of dimorphic fungi isolated in this studied. Multiple fungal organisms were cultured in six patients with a systemic fungal prescription (Appendix A).

The underutilization of indirect mycological tests such as 1,3 Beta-D-Glucan (*n* = 3; 1.7%), Cryptococcal Ag (*n* = 54; 30.5%) and *Aspergillus* Galactomannan (*n* = 46; 26.0%) was observed. Out of the 177 systemic antifungal prescriptions, 69 (39.0%) were ordered for patients who had been reviewed by the infectious disease or microbiology team. Eight adverse antifungal drug effects were noted in this study, which included deranged liver function (*n* = 5; 62.5%), drug rash (*n* = 1; 12.5%), congestive heart failure (*n* = 1; 12.5%) and monomorphic ventricular tachycardia (*n* = 1; 12.5%).

### 3.3. Antifungal Drug Utilisation Pattern

Triazoles (*n* = 107, 60.5%) were the most prescribed class of systemic antifungal in this study, followed by echinocandins (*n* = 57; 32.2%). Prescriptions of squalene epoxidase inhibitor (*n* = 7; 4.0%), polyenes (*n* = 5; 2.8%) and anti-metabolites (*n* = 1; 0.6%) were infrequent. Itraconazole (*n* = 56; 31.6%) was the most prescribed systemic antifungal in this study, followed by micafungin (*n* = 53, 29.9%) and fluconazole (*n* = 33%, 16.9%). Other antifungals combined to contribute to 35 prescriptions, which equates to a minor share of 19.8% of all systemic antifungals prescribed in PYNEH in the study period. There were four unique hospital encounters with redundant antifungal prescriptions. In all other instances, only a single antifungal agent was employed.

A total of 102 (57.6%) antifungals were administered orally, whereas 75 (42.4%) antifungals were administered intravenously. The overwhelming majority of itraconazole was prescribed orally in capsule form (55/56; 98.2%). The median duration of systemic antifungal treatment was 8 days, with an interquartile range of 5 to 17 days (Table 2).

The haematology team (*n* = 84; 47.5%) prescribed the most systemic antifungals in the study period, followed by infectious disease team (*n* = 22; 12.4%) and ICU (*n* = 16; 9.0%). The department of medicine, which includes 10 subspecialty teams (i.e., cardiology, endocrinology, diabetes and metabolism, gastroenterology and hepatology, geriatric medicine, haematology and haematological oncology, nephrology, neurology, respiratory medicine, rheumatology and infectious disease), prescribed 138 systemic antifungals as a whole; this equated to 78.0% of all systemic antifungals prescribed in PYNEH. Itraconazole was mostly prescribed by the haematology team (54/56; 96.4%) as a prophylaxis for IFD. Micafungin prescriptions were mostly ordered by the haematology team (17/53; 32.1%), ICU (11/53; 18.9%) and infectious disease team (9/53; 17.0%), whereas fluconazole prescriptions were evenly contributed to across various departments/specialties (Appendix A).

### 3.4. Indications of Antifungal Use

Antifungal prescriptions were classified according to different antifungal therapeutic strategies. There were 70 prophylactic treatments (39.5%), 55 empirical treatments (31.1%), 5 pre-emptive treatments (2.8%) and 47 targeted treatments (26.6%). All of the pre-emptive treatments were directed against possible invasive pulmonary aspergillosis (IPA). There were 18 unique episodes of IFD in the study period that accounted for 26 targeted antifungal prescriptions. The 18 unique episodes of IFD observed in this study include 12 episodes of fungemia (*Candida glabrata*, *n* = 7; *Candida tropicalis*, *n* = 1; *Candida albicans*, *n* = 1; *Candida parapsilosis*, *n* = 1; *Trichosporon faecale*, *n* = 1; *Kodamaea ohmeri*, *n* = 1), 3 episodes of probable IPA (*Aspergillus fumigatus*, *n* = 3), 1 episode of pericardial Talaromycosis (*Talaromyces marneffeii*, *n* = 1), 1 episode of epidural abscess *(Candida tropicalis*, *n* = 1; *Candida krusei*, *n* = 1) and 1 episode of cutaneous mucormycosis (*Mucor* spp.). A predominance of fungal infections caused by *Candida* spp. was observed; 26 out of 47 (55.3%) targeted antifungal treatments were made for candidiasis, which included 11 cases of invasive candidiasis (Table 3).

### 3.5. Appropriateness of Antifungal Prescription

#### 3.5.1. Overall

Only 27.7% (*n* = 49) of the systemic antifungals prescribed in PYNEH during the study period were considered to be appropriate. The indication, dosage, duration and potential for antifungal–concomitant drug interactions were deemed appropriate in 72.9% (*n* = 129), 57.1% (*n* = 101), 91.5% (*n* = 162) and 71.6% (*n* = 127) prescriptions, respectively (Table 4).

#### 3.5.2. Indication and Antifungal Choice

Indication and antifungal choice were deemed appropriate in 129 (72.9%) antifungal prescriptions. Of the 48 antifungal prescriptions with inappropriate indications, 18 targeted mycological culture that represented either colonisation or contaminations; 15 such prescriptions targeted sputum culture that grew *Candida* spp. A total of 15 systemic antifungals were prescribed in patients without any evidence suggestive of IFD, IPA or COVID-19-associated pulmonary aspergillosis (CAPA). In addition, 11 counts of an inappropriate choice of antifungal and 4 counts of redundant systemic antifungals were observed (Appendix A).

#### 3.5.3. Dosage

Antifungal dosage was considered appropriate in 101 (57.1%) prescriptions. The underwhelming prescription appropriateness in the dosage criterion is largely attributed to 52 debatable itraconazole prophylaxis prescribed at a fixed daily dose in capsule form instead of the weight-based dose in oral solution recommended by ECIL-6. A total of 19 out of 22 (86.4%) antifungal prescriptions with an inappropriate dosage were a triazole (Appendix A).

#### 3.5.4. Duration

Most (*n* = 162; 91.5%) of the systemic antifungal prescriptions were deemed to have an appropriate duration. A total of 15 antifungal prescriptions had inappropriate durations, which included an excessive duration in 12 prescriptions and an inadequate treatment duration in 3 prescriptions (Appendix A).

#### 3.5.5. Antifungal–Concomitant Drug Interactions

There were 89 antifungal prescriptions (50.3%) with a total of 169 potential significant antifungal–concomitant drug interactions in this study. Most of these interactions had a risk rating of C, “monitor therapy” (*n* = 101). However, 60 antifungal–drug interactions had a risk rating of D, “consider therapy modification”, and 8 antifungal–drug interactions had a risk rating of X, “avoid combination”. Itraconazole had the highest number of potential significant antifungal–concomitant drug interactions with 77, followed by fluconazole with 52 and voriconazole with 28.

A total of 50 (28.2%) antifungal prescriptions were deemed inappropriate due to potentially serious antifungal–concomitant drug interactions (i.e., treatment failure and adverse drug effects) that were left unattended. All of these antifungals in question belong to the triazole class, which includes itraconazole (*n* = 34), fluconazole (*n* = 9), voriconazole (*n* = 5) and posaconazole (*n* = 2). Common antifungal–concomitant drug combinations with potentially serious drug interactions (defined as having a risk rating of “D” or “X”) observed in this study include the following: itraconazole–pantoprazole (*n* = 20), itraconazole–calcium carbonate (*n* = 12), itraconazole–atorvastatin (*n* = 5), fluconazole–alprazolam (*n* = 4) and itraconazole–aprepitant (*n* = 3) (Appendix A).

#### 3.5.6. Subgroup Analysis

Triazole was the class of antifungals that was most often prescribed inappropriately. Only 16 out of 107 (15.0%) triazole prescriptions were deemed appropriate. As a quick reference, 25 out of 57 (43.9%) echinocandin prescriptions were appropriate. A majority of triazole prescriptions failed the dosage and antifungal–concomitant drug interactions assessment criteria. A 2 × 2 chi-square test of independence revealed that triazole antifungals were more likely than non-triazoles to be prescribed inappropriately (X^2^ (1, *n* = 177) = 21.9, *p* < 0.001).

Itraconazole (*n* = 35; 32.7%), micafungin (*n* = 30; 28.0%), fluconazole (*n* = 19; 17.8%) and voriconazole (*n* = 13; 12.1%) were among the most common antifungals that were prescribed inappropriately (*n* = 107). A total of 31 out of 35 (88.6%) inappropriate itraconazole prescriptions had potentially serious antifungal–concomitant drug interactions. A total of 26 out of 30 (86.7%) inappropriate micafungin prescriptions had inappropriate indications. A total of 9 out of 13 inappropriate voriconazole prescriptions had an inappropriate dosage, of which 8 lacked TDM (Table 5).

Antifungals prescribed as targeted therapy had the best appropriateness percentage in this study at 40.4%, which was followed by empirical therapy at 32.7% and pre-emptive therapy at 20%. Prophylactic antifungal prescriptions had the worst appropriateness percentage in this study at 15.7% (Table 6).

Major antifungal prescribers such as the haematology team (84 prescriptions), infectious disease team (22 prescriptions) and Intensive Care Unit (16 prescriptions) were noted to have subpar overall prescription appropriatenesses of 17.9%, 9.1% and 25%, respectively. Suboptimal antifungal prescriptions by the haematology team were mostly related to prophylactic antifungal–concomitant drug interactions (*n* = 37) and oral itraconazole prophylaxis with 200 mg daily capsules (*n* = 21). A total of 11 out of 20 (55%) inappropriate antifungal prescriptions by were given to patients with COVID-19 infection; all of these antifungal prescriptions had inappropriate indications. A total of 10 out of 12 (83.3%) inappropriate antifungal prescriptions made by the ICU targeted mycology culture that represented colonisation or contamination (Appendix A).

## 4. Discussion

Our study described the prescription patterns of systemic antifungals and evaluated the appropriateness of systemic antifungal prescription in PYNEH, a regional hospital in Hong Kong. Itraconazole (*n* = 56; 31.6%), micafungin (*n* = 53; 29.9%) and fluconazole (*n* = 33; 18.6%) were the most prescribed systemic antifungals. Despite observing a similar predominance of candidiasis among proven IFD cases previously published studies reported fluconazole, instead of itraconazole and micafungin, as the most frequently prescribed systemic antifungal in their audits [4,6,17].

The overall appropriateness of systemic antifungal prescription was 27.7%, which correlates to the lower end of the wide range of appropriateness reported by published studies [3,4,5,6]. However, direct comparison must be made cautiously as the patient characteristics, hospital settings and evaluation criteria for the appropriateness of antifungal prescriptions are not homogenous across different studies. Our study further delineates appropriateness with indications, dosage, duration and antifungal–concomitant drug interaction assessment criteria, yielding an appropriateness of 78.9% (129/177) for indications, 57.1% (101/177) for dosage, 91.5% (162/177) for duration and 71.6% (127/177) for antifungal–concomitant drug interactions. Other than the exceptionally low appropriateness percentage in the dosage assessment, which is largely due to the itraconazole capsule debacle, these figures are well within the ranges of appropriateness reported by previous studies [3,4,5,6,31,32]. The widespread inappropriate use of antifungals may be attributable to suboptimal knowledge of systemic antifungal prescription and IFD diagnosis and management among healthcare providers in general [7,8]. The dismal prescription appropriateness of triazoles (15%) may also be the product of knowledge deficiency pertaining to proper dosage, antifungal side effects, therapeutic drug monitoring, etc., and a lack of awareness of potential antifungal–drug interactions.

All prophylactic itraconazole was prescribed as oral capsules at a fixed daily dose of 200 mg in this study, which was recommended by the internal guidelines of the haematology team. This practice differs from ECIL-6, which recommends itraconazole prophylaxis with oral solution at 2.5 mg/kg Q12H [26]. There was one incident of prophylaxis failure with oral itraconazole capsule observed in this study, in which a patient with haematological malignancy and neutropenia developed and subsequently succumbed to breakthrough candidemia despite being put on prophylaxis with oral itraconazole capsules at 200 mg daily. This unfortunate incident may be attributed to an unreliable drug level due to the erratic drug absorption of oral itraconazole capsules, as warned by ECIL-6, and prohibitive antifungal–drug interactions with itraconazole–aprepitant (risk rating: X) and itraconazole–pantoprazole (risk rating: D).

Another common pitfall noted in this study is starting systemic antifungal treatment for respiratory specimen culture that grew *Candida* spp. Despite explicit recommendations printed on laboratory reports of sputum culture suggesting clinicians that “Candida organisms are not a cause of pneumonia”, 15 systemic antifungals prescribed in this study targeted respiratory specimen culture that grew *Candida* spp. This observation may reflect not only a lack of knowledge in diagnosing and managing candidiasis and/or IFD but also defensive prescription behaviour among clinicians that is also encountered in antibiotic stewardship, where fear of professional liability has led to antibiotic overuse. In the 2016 international ESCMID AntibioLegalMap survey that targeted specialists in infectious disease and clinical microbiology, the antibiotic stewards and supposed role models, it was reported that 85.0% of respondents (525/618) reported some defensive behaviour in antibiotic prescriptions [33]. The provision of guidelines and decision algorithms may be of paramount importance to change the “more is better and safer” mantra.

The results from this study highlight the widespread suboptimal use of systemic antifungals in a regional hospital in Hong Kong. Considering the ongoing threats of *Candida auris* outbreak in Hong Kong [16] and the high rate (44%) of fluconazole non-susceptible candidemia reported in Hong Kong [34], establishing an antifungal stewardship programme in public hospitals may be beneficial. Through the cessation of unnecessary antifungals and improving prescription qualities, published studies have demonstrated that antifungal stewardship programmes are beneficial. A 12-month prospective audit with non-compulsory feedback of 662 voriconazole, caspofungin and liposomal amphotericin B prescriptions in a Spain hospital led to the cessation of antifungal treatment in 8% of prescriptions, a reduction in IV voriconazole and caspofungin consumption and 11.8% reduction in antifungal expenditure [35]. Similarly, the implementation of an antifungal stewardship programme in a Greece tertiary care hospital led to a statistically significant increase in appropriate antifungal prescriptions from 47% to 76.2% (*p* = 0.01) and a statistically significant reduction in the total consumption of systemic antifungal agents from 33.4 defined daily doses/100 patient-days in the pre-interventional period to 25.5 DDDs/100 patient-days in the post-interventional period (*p* < 0.001) [9]. Inferring from the prescription patterns and appropriateness of systemic antifungals in this study, the authors opine that a selective audit of triazoles, echinocandins and liposomal amphotericin B prescriptions in the department of medicine and ICU may be a cost-effective approach to establishing an antifungal stewardship programme in the public hospitals of Hong Kong. In addition, the continued education of frontline clinicians about the intricacies of systemic antifungal prescriptions and the development of local guidelines on diagnosing and managing IFD shall play crucial roles in ensuring appropriate antifungal prescriptions in our locality.

There are some inherent limitations in this study. First, there was only a single reviewer for the assessment of prescription appropriateness. Second, this study was conducted at a single regional hospital on Hong Kong island over a relatively short period of time; the external validity of this study is yet to be determined. Third, due to the nature of this study being a retrospective electronic chart review, missing clinical information or insufficient documentation may confound results and influence the conclusion. Despite its fundamental constraints in study design, this study provides an in-depth review regarding the patterns and appropriateness of systemic antifungal prescriptions in a regional hospital in Hong Kong. We look forward to future descriptive studies on knowledge, attitude and practice regarding antifungal prescriptions and IFD in Hong Kong, as well as quasi-experiments of antifungal stewardship programme implementation in local public hospital settings to guide forthcoming efforts to improve the quality of antifungal prescriptions in Hong Kong.

## 5. Conclusions

Inappropriate systemic antifungal prescription is not uncommon in Hong Kong. The overall appropriateness of systemic antifungal prescription in this study is 27.7%. Itraconazole, micafungin and fluconazole were the most prescribed antifungals. Triazoles were the most prescribed class of systemic antifungal; they were also significantly more likely than non-triazoles to be prescribed inappropriately. The common pitfalls of antifungal prescriptions encountered in this study include (i) starting systemic antifungal treatment for sputum culture that grew *Candida* spp., (ii) debatable prophylaxis with itraconazole capsules, (iii) overlooking potentially serious antifungal–drug interactions.

The findings of this study suggest that establishing an antifungal stewardship programme that targets triazoles, echinocandins and liposomal amphotericin B prescriptions in the department of medicine and ICU may be a cost-effective approach to improving prescription quality in Hong Kong. In addition, the development of local guidelines for diagnosing and managing IFD and educating frontline doctors on the proper use of systemic antifungals are of paramount importance.

## Figures and Tables

**Table 1 antibiotics-14-00556-t001:** Underlying conditions and other factors predisposing to IFD (*n* = 177).

Underlying Condition ^†^	*n*	%
Haematological malignancy	95	53.7
Diabetes mellitus	24	13.6
Haematopoietic stem cell transplant	4	2.3
Solid organ transplant	3	1.7
Solid tumour	2	1.1
HIV/AIDS	1	0.6
Other immunocompromised conditions ^‡^	3	1.7
**Other predisposing factor** ^†,§^		
Neutropenia	78	44.1
Chemotherapy	86	48.6
Corticosteroid therapy	49	27.7
Other immunosuppressants ^¶^	24	13.6
COVID-19	28	15.8
Central venous catheter	27	15.3
Total parenteral nutrition	8	4.5
Intra-abdominal infection	7	4.0
Chronic obstructive lung disease/chronic lung disease	6	3.4
End-stage renal failure requiring renal replacement therapy	5	2.8
Premature/LBW	5	2.8

^†^ A patient could have more than one underlying condition or predisposing factor. ^‡^ Includes severe G6PD deficiency and interferon gamma autoantibody. ^§^ Predisposing factors in one patient may have changed within the study period. ^¶^ Immunosuppressants other than corticosteroids and chemotherapeutic agents.

**Table 2 antibiotics-14-00556-t002:** Systemic antifungal utilisation in PYNEH from 1 May to 31 July 2023.

Antifungal (Route, Dosage Form)	*n*	%
**Triazoles**	**107**	**60.5**
Fluconazole (IV)	11	6.2
Fluconazole (PO)	22	12.4
Itraconazole (PO Capsule)	55	31.1
Itraconazole (PO Solution)	1	0.6
Voriconazole (IV)	2	1.1
Voriconazole (PO)	12	6.8
Isavuconazole (PO)	1	0.6
Posaconazole (PO)	3	1.7
**Polyenes**	**5**	**2.8**
Amphotericin B Deoxycholate (IV)	1	0.6
Liposomal Amphotericin B (IV)	4	2.3
**Echinocandins**	**57**	**32.2**
Micafungin	53	29.9
Anidulafungin	4	2.3
**Squalene Epoxidase Inhibitor**	**7**	**4.0**
Terbinafine	7	4.0
**Anti-Metabolite**	**1**	**0.6**
Flucytosine	1	0.6

IV: intravenous; PO: per os.

**Table 3 antibiotics-14-00556-t003:** Antifungal therapeutic strategies (*n* = 177 prescriptions).

	*n*	%
**Prophylactic treatment**	**70**	**39.5**
**Empirical**	**55**	**31.1**
**Pre-emptive**	**5**	**2.8**
Possible IPA	5	2.8
**Targeted**	**47**	**26.6**
Candidemia	10	21.3
*Candida* CAPD peritonitis	3	6.4
*Candida* CNS infection	1	2.1
Oropharyngeal candidiasis	7	14.9
Oesophageal candidiasis	3	6.4
*Candida* vulvovaginitis	1	2.1
Candiduria	1	2.1
Probable IPA	3	6.4
Dermatophytosis	3	6.4
Onychomycosis	3	6.4
Cryptococcal meningitis	3	6.4
Talaromycosis	2	4.3
*Penicillium* CAPD peritonitis	2	4.3
*Trichosporon faecale* fungaemia	2	4.3
*Kodamaea ohmeri* fungaemia	2	4.3
Cutaneous mucormycosis	1	2.1

**Table 4 antibiotics-14-00556-t004:** Appropriateness of systemic antifungal prescription in this study (*n* = 177).

Assessment Criteria	Appropriate (%)	Inappropriate (%)	Debatable (%)
**Overall**	**49 (27.7%)**	**107 (60.5%)**	**21 (11.9%)**
Indication	129 (72.9%)	48 (27.1%)	-
Dosage	101 (57.1%)	22 (12.4%)	54 (30.5%)
Duration	162 (91.5%)	15 (8.5%)	-
Antifungal–drug interaction	127 (71.6%)	50 (28.2%)	-

**Table 5 antibiotics-14-00556-t005:** Appropriateness of different antifungal drugs prescribed in this study.

Antifungal	Appropriate (%)	Inappropriate (%)	Debatable (%)	Subtotal
**Triazoles**	**16 (15.0%)**	**70 (65.4%)**	**21 (19.6%)**	**107**
Fluconazole	14 (42.4%)	19 (57.6%)	-	33
Itraconazole	-	35 (62.5%)	21 (37.5%)	56
Voriconazole	1 (7.1%)	13 (92.9%)	-	14
Isavuconazole	-	1 (100%)	-	1
Posaconazole	1 (33.3%)	2 (66.7%)	-	3
**Polyenes**	**4 (80%)**	**1 (20%)**	**-**	**5**
Amphotericin B Deoxycholate	1 (100%)	-	-	1
Liposomal Amphotericin B	3 (75%)	1 (25%)	-	4
**Echinocandins**	**25 (43.9%)**	**32 (56.1%)**	**-**	**57**
Micafungin	23 (43.4%)	30 (56.6%)	-	53
Anidulafungin	2 (50%)	2 (50%)	-	4
**Squalene Epoxidase Inhibitor**	**4 (57.1%)**	**3 (42.9%)**	**-**	**7**
Terbinafine	4 (57.1%)	3 (42.9%)	-	7
**Anti-Metabolite**	**-**	**1 (100%)**	**-**	**1**
Flucytosine	-	1 (100%)	-	1
**Total**	**49 (27.7%)**	**107 (60.5%)**	**21 (11.9%)**	**177**

**Table 6 antibiotics-14-00556-t006:** Appropriateness of antifungal prescriptions by different therapeutic strategies.

Treatment Strategies	Appropriate (%)	Inappropriate (%)	Debatable (%)	Total
Empirical	18 (32.7%)	37 (67.3%)	-	55
Pre-emptive	1 (20%)	4 (80%)	-	5
Targeted	19 (40.4%)	28 (59.6%)	-	47
Prophylaxis	11 (15.7%)	38 (54.3%)	21 (30%)	70
**Total**	**49 (27.7%)**	**107 (60.5%)**	**21 (11.9%)**	**177**

## Data Availability

The original contributions presented in this study are included in the article/Appendix A. Further inquiries can be directed to the corresponding author(s).

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
