# Peer review of "The Patterns and Appropriateness of Systemic Antifungal Prescriptions in a Regional Hospital in Hong Kong"

_antibiotics, 2025, doi:10.3390/antibiotics14060556_

Round 1

Reviewer 1 Report

Comments and Suggestions for Authors

The present retrospective study reports the usage pattern of antifungals in a single hospital in Hong Kong. The study seems to be conducted well and authors have provided the relevant data comprehensively. Few comments from my side are:

The study highlights a widespread sub-optimal use of systemic anti-fungals in a single hospital in Hong Kong. Based on the observations, what specific recommendations do the authors propose to improve rational use of these agents in their setting?

Does the hospital have a functioning antimicrobial/ antibiotic stewardship programme?

In discussion, a brief mention of the antifungal stewardship measures has been given in lines 344-347, would suggest to add some more details in this context.

Comments on the Quality of English Language

Minor editing in English language required.

Author Response

Thank you for your comments!!

1. The study highlights a widespread sub-optimal use of systemic antifungals in a single hospital in Hong Kong. Based on the observations, what specific recommendations do the authors propose to improve rational use of these agents in their setting?

We propose the following measures to improve rational use of systemic antifungals in Hong Kong:

  • Establish an antifungal stewardship programme
    • Selective audit triazoles, echinocandins and liposomal amphotericin B prescriptions in department of medicine and ICU may be a cost-effective approach.
  • Educate frontline clinicians about the intricacies of systemic antifungal prescriptions
  • Develop of local guidelines on diagnosing and managing invasive fungal disease (IFD)

2. Does the hospital have a functioning antimicrobial/ antibiotic stewardship programme?

PYNEH currently has an antibiotic stewardship programme that offer prospective audits with immediate concurrent feedbacks on big gun antibiotics, which includes piperacillin-tazobactam, cefoperazone-sulbactam, cefepime, ertapenem, imipenem-cilastin, meropenem and vancomycin.

3. In discussion, a brief mention of the antifungal stewardship measures has been given in lines 344-347, would suggest to add some more details in this context.

We have expanded the discussion regarding the positive impact of antifungal stewardship programmes in published studies in line 399-416

Reviewer 2 Report

Comments and Suggestions for Authors

Major points

- the introduction should be expanded as it does not provide a clear overview of the examined problems; several points should be added in the introduction: risk factors for IFD, prevalence of IFD by type of infection and pathogen, expand the previous data regarding the appropriateness of systemic antifungal prescriptions (ref3-6), expand the problematic consequences regarding overuse and misuse of systemic antifungals; Furthermore, is there evidence indicating whether overuse or misuse is the more prevalent issue in the administration of systemic antifungals?

- line 97 “Indication, dosage, duration and potential antifungal-concomitant drug interactions were evaluated by a single reviewer.”; specify the reviewer that evaluated these aspects; was guidance from the clinical microbiology or infectious disease department sought or required regarding these aspects?

- line 115 – how were data anonymised?

- Table 1 – several changes are required to table 1; reformat the table to increase the readability of the data; furthermore, please include an additional table presenting the predisposing factors in relation to the patients rather than the antifungal treatments, as some patients received more than one antifungal agent – this would provide a clearer overview of the data

- section 3.2 – specify the number of patients with more than one identified fungal pathogen; provide a minimal description of the identification methods for the mentioned pathogens

- section 3.3 – specify the number of patients with more than one administered antifungal

- section 3.4 – several aspects of this section are not entirely clear; this section states that there were 18 unique episodes of IFD while the previous section specify that a positive fungal culture was reported in 55 systemic antifungal prescriptions; clearly specify the number of unique IFD and the respective pathogens involved in each case

- line 295 – this is an interesting finding; most often fluconazole is indeed the first choice of treatment in the case of both superficial and superficial fungal infections; are there any possible explanations for this finding? (e.g., local or hospital guidelines, availability, price or others)

- line 318 – this is also an interesting finding; are there any particular reasons for which the haematology team implemented this specific dosage in the internal guidelines?

- line 333 – are all cultured microorganisms reported by the microbiology laboratory, even if their presence most likely signifies colonisation? (e.g., Candida from sputum samples)

- line 336 – defensive prescription might indeed be a significant factor in this case; please expand this part of the discussion due to its importance

- add a conclusions section to the manuscript which clearly highlights the most important findings of the study

- Supplementary Table 1 – if the study focused on invasive fungal infections, why were “superficial candidiasis and dermatophytosis” included in the targeted antifungal therapy definition?

- Supplementary Table 8 – this table has decreased readability due to the absence of clearly defined rows

Minor points

- line 112 “UpToDate Lexidrug online software” – also add the respective link for the online software

- line 185 – clarify Department of medicine; does it refer to all medical specialties as opposed to surgical ones?

- Supplementary tables should be correctly annotated as such (Scheme is not appropriate in this context)

Comments on the Quality of English Language

English revision of the manuscript is required

Author Response

Thank you for your comments!

Major points

1. The introduction should be expanded as it does not provide a clear overview of the examined problems; several points should be added in the introduction: risk factors for IFD, prevalence of IFD by type of infection and pathogen, expand the previous data regarding the appropriateness of systemic antifungal prescriptions (ref3-6), expand the problematic consequences regarding overuse and misuse of systemic antifungals; Furthermore, is there evidence indicating whether overuse or misuse is the more prevalent issue in the administration of systemic antifungals?

More background information was added to give a more comprehensive overview regarding IFD, antifungal overuse/misuse and the challenges we face in ensure appropriate antifungal prescriptions

2. line 97 “Indication, dosage, duration and potential antifungal-concomitant drug interactions were evaluated by a single reviewer.”; specify the reviewer that evaluated these aspects; was guidance from the clinical microbiology or infectious disease department sought or required regarding these aspects?

Dr. Ryan Leung (author), who is a member of the microbiology team, is the single reviewer that evaluated the appropriateness of systemic antifungal prescriptions. The study was carried out under the auspices of the microbiology team.

3. line 115 – how were data anonymised?

Each prescription order was given a unique code, of which relevant clinical data was retrieved from patient electronic records. 

4. Table 1 – several changes are required to table 1; reformat the table to increase the readability of the data; furthermore, please include an additional table presenting the predisposing factors in relation to the patients rather than the antifungal treatments, as some patients received more than one antifungal agent – this would provide a clearer overview of the data

Due to evolving clinical picture and patient condition in the same patient throughout the study period (e.g. development/resolution of neutropenia, administration of chemotherapy, insertion/removal of central venous catheter) that prompted the initiation or cessation of antifungals, the authors consider that presenting the predisposing factors of a certain patient in relation to each systemic antifungal prescription to be more appropriate to describe prescription patterns and evaluate appropriateness in this study.

Tables format has been amended.

5. Section 3.2 – specify the number of patients with more than one identified fungal pathogen; provide a minimal description of the identification methods for the mentioned pathogens

Multiple fungal organisms were isolated in 6 systemic fungal prescriptions.

Microscopic and colony morphology, as well Matrix-Assisted Laser Desorption/Ionization–Time Of Flight Mass Spectrometry (MALDI-TOF-MS) were employed to identify the isolated fungal organisms,

6. section 3.3 – specify the number of patients with more than one administered antifungal

There were 4 unique hospital encounters with redundant antifungal prescriptions (i.e. multiple antifungal prescriptions). In all other instances, only single antifungal agent was employed.

7. section 3.4 – several aspects of this section are not entirely clear; this section states that there were 18 unique episodes of IFD while the previous section specify that a positive fungal culture was reported in 55 systemic antifungal prescriptions; clearly specify the number of unique IFD and the respective pathogens involved in each case

The 18 unique episodes of IFD observed in this study include 12 episodes of fungemia (Candida glabrata, N = 7, Candida tropicalis, N = 1; Candida albicans, N = 1; Candida parapsilosis, N = 1; Trichosporon faecale, N = 1; Kodamaea ohmeri, N = 1), 3 episodes of probable IPA (Aspergillus fumigatus, N = 3), 1 episode of pericardial Talaromycosis (talaromyces marneffeii, N = 1), 1 episode of epidural abscess (Candida tropicalis, N=1; candida krusei, N=1), 1 episode of cutaneous mucormycosis (mucor spp.)

8. line 295 – this is an interesting finding; most often fluconazole is indeed the first choice of treatment in the case of both superficial and superficial fungal infections; are there any possible explanations for this finding? (e.g., local or hospital guidelines, availability, price or others)

A significant proportion (39.5%; N=70) of systemic antifungal were prescribed as IFD prophylaxis in this study. The majority of itraconazole was prescribed by haematology team (54/56; 96.4%) as a prophylaxis for IFD, which is driven by the internal guidelines per haematology team and physicians’ preference for mold coverage in IFD prophylaxis.

The high usage rate of micafungin may be attributable to the common occurrence of non-albicans Candida spp. isolated from clinical specimen (25/44; 56.8%) as well as better side effect and drug-drug interaction profile compared to fluconazole.

9. line 318 – this is also an interesting finding; are there any particular reasons for which the haematology team implemented this specific dosage in the internal guidelines?

There is no rationale provided by haematology team regarding their preference of fixed daily dose of itraconazole capsule over the suggested weight based dosing of itraconazole solution by ECIL-6 guidelines.

10. line 333 – are all cultured microorganisms reported by the microbiology laboratory, even if their presence most likely signifies colonisation? (e.g., Candida from sputum samples)

Our laboratory only report Candida spp. isolate from sputum specimen with heavy growth. In addition, the comment “Candida organisms are not a cause of pneumonia” will be added to deter clinicians from initiating antifungal treatment.

11. line 336 – defensive prescription might indeed be a significant factor in this case; please expand this part of the discussion due to its importance

We have expanded the discussion regarding the defensive behavior in antibiotic prescription and antibiotic stewardship in line 384-393.

12. add a conclusions section to the manuscript which clearly highlights the most important findings of the study

A conclusion section was added

13. Supplementary Table 1 – if the study focused on invasive fungal infections, why were “superficial candidiasis and dermatophytosis” included in the targeted antifungal therapy definition?

Describing systemic antifungal prescriptions patterns and evaluate the prescription appropriateness are the key objectives of this study. A significant proportion of systemic antifungal uses are indeed targeted therapy towards a clinical diagnosis of superficial candidiasis and dermatophytosis and thus these presciptions orders were included in this study.

14. Supplementary Table 8 – this table has decreased readability due to the absence of clearly defined rows

Table format has been amended

Minor points

1. line 112 “UpToDate Lexidrug online software” – also add the respective link for the online software

Repsective link for the online software was added

2. line 185 – clarify Department of medicine; does it refer to all medical specialties as opposed to surgical ones?

Department of medicine includes 10 subspecialty teams (i.e. cardiology, endocrinology, diabetes & metabolism, gastroenterology & hepatology, geriatric medicine, haematology & haematological oncology, nephrology, neurology, respiratory medicine, rheumatology and infectious disease).

3. Supplementary tables should be correctly annotated as such (Scheme is not appropriate in this context)

Table format has been amended 

Round 2

Reviewer 2 Report

Comments and Suggestions for Authors

In my opinion, no further improvements to the manuscript are required